# MathScape: Evaluating MLLMs in multimodal Math Scenarios through a Hierarchical Benchmark

## Abstract

With the development of Multimodal Large Language Models (MLLMs), the evaluation of multimodal models in the context of mathematical problems has become a valuable research field. Multimodal visual-textual mathematical reasoning serves as a critical indicator for evaluating the comprehension and complex multi-step quantitative reasoning abilities of MLLMs. However, previous multimodal math benchmarks have not sufficiently integrated visual and textual information. To address this gap, we proposed MathScape, a new benchmark that emphasizes the understanding and application of combined visual and textual information. MathScape is designed to evaluate photo-based math problem scenarios, assessing the theoretical understanding and application ability of MLLMs through a categorical hierarchical approach. We conduct a multi-dimensional evaluation on 11 advanced MLLMs, revealing that our benchmark is challenging even for the most sophisticated models. By analyzing the evaluation results, we identify the limitations of MLLMs, offering valuable insights for enhancing model performance. The code is made available `https://anonymous.4open.science/r/MathScape-8742`.

## 1 Introduction

Large language models (LLMs) have demonstrated exceptional performance across diverse tasks spanning myriad domains OpenAI (2023a); Touvron et al. (2023). Based on LLMs, MLLMs Zhao et al. (2023); Wu et al. (2023); Bai et al. (2024) also show strong understanding ability among different modalities Liu et al. (2023b); Bai et al. (2023b). Among Multimodal Large Language Models (MLLMs), Vision Language Large Models (VLLMs) have demonstrated competitive performance in traditional multimodal tasks, including image classification Chen et al. (2024), image understanding Li et al. (2023b;c), and image captioning Bai et al. (2023b). Furthermore, their advanced language understanding capabilities contribute to strong performance in text-rich tasks, such as visual question answering Liu et al. (2023b;a) and image-text retrieval Chen et al. (2024). Recently, VLLMs have also shown significant progress in solving mathematical problems. Therefore, comprehensive benchmarks are essential to evaluate the mathematical abilities of VLLMs. Although several benchmarks, such as MATH-

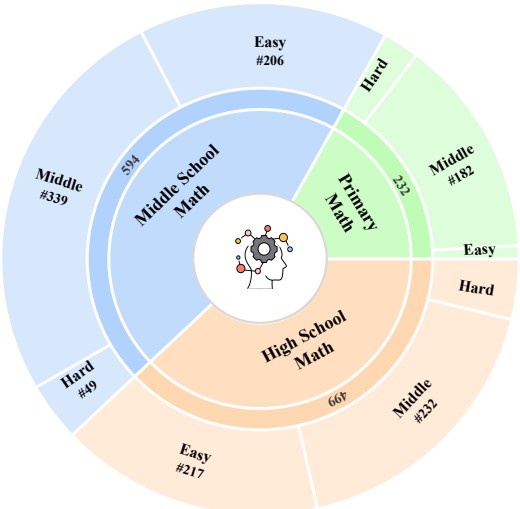

Figure 1: MathScape offers a comprehensive collection of math problems from primary school to high school. The problems range in difficulty from easy to difficult, catering to various levels of evaluation.

V (Wang et al., 2024a), MathVerse (Zhang et al., 2024a), and MathVista (Lu et al., 2023b), have been developed to assess the mathematical capabilities of VLLMs. They primarily focus on a combination of text math problems and image figures. Also, they only use simple metrics and lack effective evaluation for complex or extended responses. Consequently, they face two key challenges:

**C1. Insufficient Real-World Data.** In previous datasets like MATH-V (Wang et al., 2024a), Math-Verse (Zhang et al., 2024a), and MathVista (Lu et al., 2023b), the mathematical description was typically provided as text input, while the image contained only figures. This approach doesn't align well with real-world scenarios, where both the mathematical description and figures are captured together in a single image.

**C2. Absence of Effective Evaluation Metrics.** In previous datasets Wang et al. (2024a); Zhang et al. (2024a); Lu et al. (2023b), the evaluation was limited to short answers, lacking the ability to assess long-form responses.

To address these issues, we implement a three-step pipeline for constructing a real-world math image dataset. As illustrated in Figure 3, the process begins by converting math documents into images, as shown in Figure 2. Next, we capture photos and screenshots to build the dataset. Finally, we perform a thorough review and knowledge classification to ensure the dataset's high quality. For evaluation, we design a two-step pipeline specifically for assessing longer math problems. First, we use LLMs to extract answers for each subproblem. Then, we employ LLMs as evaluators to assess the correctness of each solution. With the data construction and evaluation pipeline, we constructed MathScape, a new multimodal dataset that combines photos of real-world math problems with their correct answers.

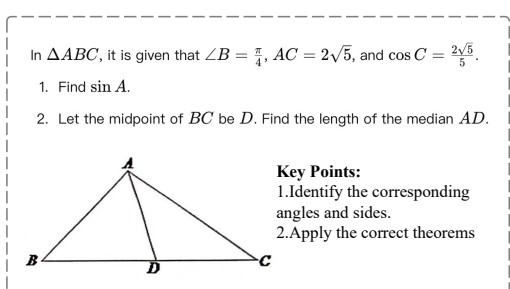

Figure 2: An example problem from MathScape. Examples in MathScape are represented by images taken by humans, ensuring a more realistic scenario. Each example will contain a correct answer.

The core contributions are summarized as follows:

- **New Perspective:** To the best of our knowledge, we are the first to construct images that combine both figures and mathematical text descriptions, closely mirroring real-world scenarios.
- **New Method:** We propose a novel three-step dataset construction pipeline, as illustrated in Figure 3. Additionally, we introduce a new two-step evaluation method specifically designed for assessing long answers.
- **New Benchmark:** We present MathScape, a new multimodal mathematical dataset that spans various difficulty levels, question types, and knowledge areas, providing a comprehensive tool to evaluate the mathematical capabilities of MLLMs. Moreover, MathScape is entirely original, consisting of previously unreleased multimodal mathematical data.

## 2 RELATED WORK

In the field of MLLMs, the benchmark for multimodal mathematical reasoning capability represents a significant and novel research direction. Mathematical reasoning is a crucial indicator for evaluating the ability of LLMs to perform complex, multi-step reasoning and quantitative analysis within visual contexts. Below, we highlight some relevant work and the latest developments in this area.

### 2.1 BENCHMARK FOR MATHEMATICAL EVALUATION

Recent research has seen significant advancements in mathematical reasoning benchmarks aimed at evaluating mathematical abilities. In this summary, we review both pure text and multimodal math benchmarks.

**Pure Text Benchmarks** GSM8K Cobbe et al. (2021) is a dataset from OpenAI that includes 8.5K high-quality elementary school math word problems, each requiring 2 to 8 steps to solve. These problems primarily involve basic arithmetic operations such as addition, subtraction, multiplication, and division. MATH Hendrycks et al. (2021) offers a dataset of 12,500 problems sourced from high school math competitions. SuperCLUE-Math Xu et al. (2024) is a Chinese benchmark for multi-step reasoning in mathematics, containing over 2,000 problems that require multi-step reasoning and offer natural language solutions. MathBench Liu et al. (2024b) includes 3,709 math problems ranging from basic arithmetic to college-level questions, covering multiple difficulty levels.

All these benchmarks focus exclusively on text-based mathematical tasks. They are designed to evaluate the mathematical capabilities of LLMs through specialized problem sets.

**Multimodal Benchmarks** With the rapid advancement of MLLMs, several high-quality benchmarks have emerged to evaluate mathematical problem-solving in visual contexts. MathVista Lu et al. (2023b) focuses on visual math QA tasks, assessing model performance across various math domains, such as arithmetic and algebra, using visual scenarios. MATH-V (Wang et al., 2024a) is another benchmark that targets multimodal mathematical understanding, with questions primarily sourced from math competitions. MathVerse Zhang et al. (2024a) evaluates MLLMs' comprehension of visual diagrams using CoT (Chain of Thought) strategies on 2,612 multimodal math problems. CMMU He et al. (2024) is a large-scale Chinese benchmark for multi-disciplinary, multimodal understanding, featuring questions from college exams and textbooks.

Compared to these existing multimodal mathematical benchmarks, which often have limitations in question length, complexity, and openness to model answers, our MathScape benchmark is designed to be longer and more open-ended.

## 2.2 MLLMs FOR MATHEMATICS

**Commonly Used VLLMs** The integration of visual knowledge into LLMs has become a pivotal area of research due to the rapid advancements in LLMs. VLLMs combine vision information from vision encoders with LLMs, thus enabling these models to process and interpret visual inputs for various visual tasks Liu et al. (2023c); Zhang et al. (2022); Li et al. (2022b) with enhanced accuracy and efficiency. Pioneering frameworks like CLIP Radford et al. (2021) leverage contrastive learning on expansive image-caption datasets to align modalities, forming the groundwork for cross-modal comprehension. Various adapters Liu et al. (2023b;a); Li et al. (2023b; 2022a); Jian et al. (2023); Lu et al. (2023a) are introduced to further integrate different modalities. For example, LLaVA Liu et al. (2023b;a) employs a straightforward MLP to inject the vision information into LLMs. Whereas more complex implementations like the Q-Former in BLIP Li et al. (2022a; 2023b) utilize cross-attention to enhance modality integration.

Recent studies Wang et al. (2024b); Chen et al. (2023); Liu et al. (2023b;a); Li et al. (2023a) aims to boost VLLM performance by focusing on the quality of both pre-training and fine-tuning datasets. Models like LLaVA Liu et al. (2023b;a) and ShareGPT4V Chen et al. (2023) have shown remarkable advancements in understanding and following complex instructions through instruction tuning.

**VLLMs Designed for Math Problems** In real-world applications, vision inputs are commonly used to present mathematical problems for models to solve. As a result, it is crucial for Vision-Language Large Models (VLLMs) to demonstrate strong mathematical capabilities. Meidani et al. Meidani et al. (2023) pioneered the use of symbolic data to train a Vision-Language Model (VLM) with mathematical proficiency. Building on this work, UniMath Liang et al. (2023) combined vision, table, and text encoders with LLMs, achieving state-of-the-art (SOTA) performance at the time. Additionally, Huang et al. Huang et al. (2024) succeeded in solving algebraic problems that involved geometric diagrams.

Another noteworthy line of research involves using LLMs to tackle geometric problems. G-LLaVA Gao et al. (2023) fine-tuned LLaVA Liu et al. (2023b) with geometric data, reaching SOTA performance in geometry. Subsequently, MAViS Zhang et al. (2024b) and EAGLE Li et al. (2024) achieved SOTA results by introducing math-specific encoders and amassing large amounts of mathematical data.

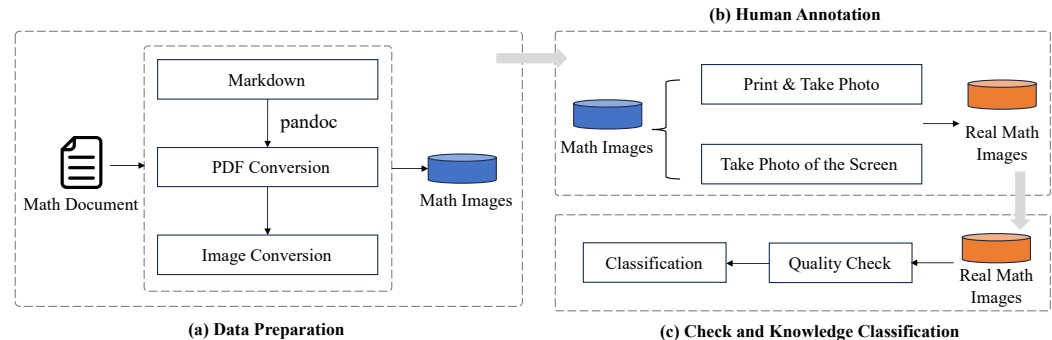

Figure 3: MathScape process pipeline.

## 3 METHODOLOGY

We begin by introducing the construction pipeline of MathScape in Section 3.1. Next, we present the multidimensional evaluation approach in Section 3.2. In Section 3.3, we detail the two-step answer evaluation method. Finally, we summarize the dataset statistics in Section 3.4.

### 3.1 CONSTRUCTION OF MATHSCAPE

**Data Preparation** The data preparation module consists of three steps, as shown in Figure 3(a). First, we collected a large number of mathematics questions from elementary, junior high, and senior high school exams and homework as the evaluation sample. We gathered a total of 1,325 image mathematics questions. Next, the question documents were converted to PDF format using Pandoc and subsequently transformed into images for further use.

**Data Annotation** As illustrated in Figure 3(b), the images are then transformed to closely align with real-world scenarios by capturing photos of printed images and screen displays.

**Data Check and Knowledge Classification** After constructing the dataset, we perform a double-check and knowledge-based classification to ensure its high quality. As illustrated in Figure 3(c), we rigorously review the dataset to ensure that both the textual and graphical inputs are clear and accurate. Once data quality is verified, we categorize the data according to knowledge points.

### 3.2 MULTIDIMENSIONAL EVALUATION

To comprehensively evaluate the performance of VLLMs, we designed multiple dimensions to classify and assess their mathematical abilities across various categories. The classification types we used are as follows:

**Question Types:** We first categorized the test questions into different types, such as multiple-choice, fill-in-the-blank (Solution), and proof questions, to examine the model's performance across various question formats.

**Knowledge Points:** We also classified the questions based on mathematical knowledge areas, including algebra, geometry, probability, and statistics, to assess the model's proficiency in different domains of mathematics.

**Educational Stages:** Additionally, the questions were divided according to the educational stage—primary school, middle school, and high school—to evaluate the model's adaptability and accuracy at different levels of education.

### 3.3 EVALUATION METHOD

We utilize a two-step evaluation process to effectively score long answers.

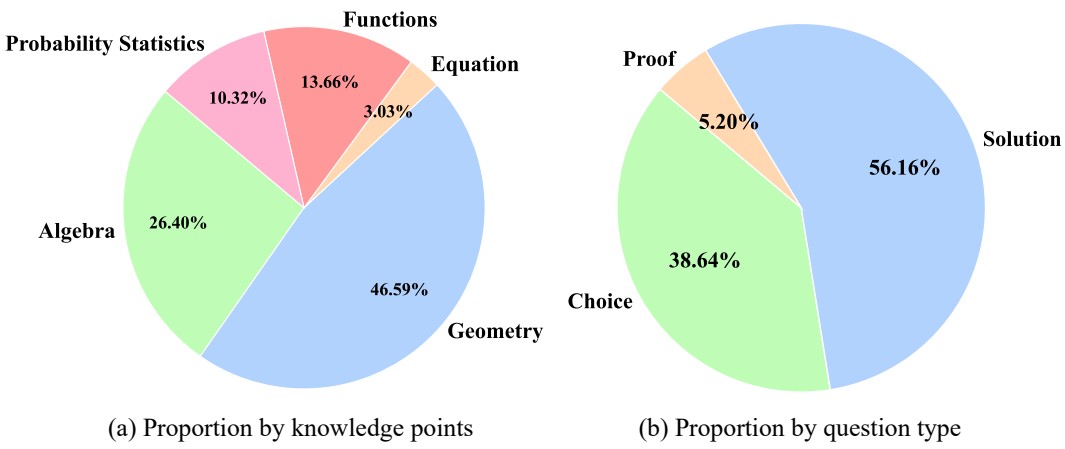

(a) Proportion by knowledge points          (b) Proportion by question type

Figure 4: Proportion Figure

**Answer Segmentation:** As illustrated in Figure 13, we prompt the LLMs to decompose a lengthy answer into multiple sub-answers, each one focusing on a specific aspect of the problem. This segmentation ensures that the complex answer is broken down into manageable components, making it easier to evaluate the correctness and relevance of each part. By isolating sub-problems within the overall solution, we can achieve a more granular analysis of the model's performance.

**Sub-Answer Scoring:** After segmenting the long answer, we employ the prompt depicted in Figure 14 to automatically score each sub-answer individually. This method allows us to evaluate the accuracy of each component independently, ensuring that the final score reflects the model's ability to handle various aspects of the problem comprehensively. By scoring sub-answers separately, we can identify specific areas where the model excels or struggles, providing deeper insights into its strengths and weaknesses.

## 3.4 DATASET STATISTICS

In this section, we provide a summary of the statistics for our MathScape dataset. The dataset primarily consists of Chinese image-text problems, along with question labels, attribute information, problem-solving processes, and standard reference answers. Detailed statistics are presented in Figure 4.

As shown in Figure 4(a), our dataset thoughtfully incorporates the characteristics of multimodal image-text questions. A significant portion of the questions are geometric, which often require the integration of images for effective problem-solving. In contrast, topics like equations and inequalities are less represented, aligning more closely with the specific demands of multimodal assessment.

Figure 4(b) illustrates that our dataset primarily includes solution questions and multiple-choice questions, with fewer proof questions. This distribution indicates that our dataset is designed to challenge models with diverse question types, while still reflecting the real-world emphasis on practical problem-solving.

Overall, our dataset contains a total of 1,325 images, providing a robust resource for evaluating the mathematical reasoning capabilities of MLLMs.

## 4 EXPERIMENTS AND ANALYSIS

In this section, we utilize multiple state-of-the-art (SOTA) models and test their performance on the MathScape benchmark.

## 4.1 EXPERIMENTAL SETUPS

**Models.** In our evaluation of multimodal LLMs, we focused on both open-source and closed-source models that rank among the top performers on major multimodal LLM leaderboards. This included 11 different types of VLLMs, with a particular emphasis on analyzing the results and performance of the leading models. For Closed-source models, we evaluate GPT4 OpenAI (2023b), GeminiPro Reid et al. (2024), Claude-3-Opus, Baichuan-VL Yang et al. (2023), Qwen-Max Bai et al. (2023a), Qwn-Plus Bai et al. (2023a), GLM4V. For Open-source models, we evaluate Deepseek-VLLu et al. (2024), LLaVALiu et al. (2024a), YiYoung et al. (2024).

**Settings.** We conduct all model inferences in a zero-shot setting, using the same configuration for each official model. Instead of the Chain of Thought (CoT) technique, we use a custom prompt to guide the model in producing the problem-solving process and final answer, as shown in Figure 13. The settings include a max token limit of 2048, top-k of 5, a temperature of 0.3, and a repetition penalty of 1.05. All experiments are run on NVIDIA H100 GPUs.

## 4.2 PERFORMANCE OF VARIOUS MODELS

In this section, we present the performance of commonly used MLLMs on our benchmark. We analyze the results from the perspectives of Question Types, Knowledge Points, and Educational Stages:

**Question Types** As shown in Table 1, GPT-4V and GPT-4-turbo exhibits the highest accuracy across all question types, with an average of 34.96%, followed by GPT-4-Turbo Vision at 33.92%. While Yi-VL-34B and DeepSeek-V2 achieve good performance among open-source models. We can see the performance of closed-source models achieved better performance than open-source models. The table shows that models generally perform better on proof questions compared to multiple-choice and solution questions. This suggests that the structured format and clear information in proof questions make them easier for models to handle, while solution questions, which require complex, multi-step reasoning, pose more of a challenge.

Table 1: **Accuracy scores comparison of models on different question types**

| Model | Average | Choice | Solution | Proof |
|---|---|---|---|---|
| Closed-source Models | | | | |
| GPT-4V | **34.96** | **35.75** | **31.72** | 28.33 |
| GPT-4-turbo | 33.92 | 29.85 | 31.58 | **56.62** |
| Claude-3-Opus | 28.79 | 29.3 | 20.85 | 50.00 |
| Gemini-Pro | 21.37 | 12.62 | 16.16 | 37.50 |
| Baichuan-VL | 30.00 | 26.38 | 25.83 | 45.97 |
| Qwen-VL-Max | 27.83 | 23.97 | 22.17 | 34.85 |
| Qwen-VL-Plus | 15.60 | 19.46 | 12.48 | 35.19 |
| GLM4V | 12.26 | 11.54 | 7.31 | 26.28 |
| Open-source Models | | | | |
| Yi-VL-34B | **18.36** | **19.01** | 9.98 | 33.33 |
| DeepSeek-V2 | 15.66 | 12.75 | **10.60** | **37.69** |
| LLaVA-1.6-7B | 12.35 | 11.31 | 6.24 | 13.43 |

**Knowledge Points** Table 2 shows the answer accuracy of the models in different knowledge points. GPT-4V and GPT-4-turbo consistently outperform other models in areas like algebra, equations and inequalities, functions, and probability and statistics. Most models show balanced performance across different knowledge areas, but there are exceptions, such as LLaVA-1.6, which does well in equations and inequalities but struggles with functions.

Overall, closed-source models are more accurate than open-source ones, with GPT-4V and GPT-4-turbo leading in many categories.

**Educational Stages** Table 3 presents the performance of open-source and closed-source models on MathScape at the elementary, middle, and high school levels. At the elementary and middle levels, the models perform similarly. However, when the difficulty increases to the high school level, we observe a significant drop in accuracy. Some models show an extreme decrease in performance between the middle and high school benchmarks. For instance, Gemini-Pro has an average accuracy of 25.79% at the elementary level, but this sharply declines to just 10.22% at the high school level. This suggests that high school-level math poses significant challenges for LLMs.

Overall, our evaluation shows that closed-source models, particularly GPT-4V and GPT-4-turbo, consistently outperform open-source models across various question types, knowledge points, and

Table 2: **Accuracy scores comparison of Models on different knowledge points**

| Model | Algebraic | Geometric | Equations | Functions | Probability Statistics |
|-------|-----------|-----------|-----------|-----------|------------------------|
| *Closed-source Models* | | | | | |
| GPT-4V | **39.05** | 27.90 | 29.73 | **34.14** | **41.31** |
| GPT4-turbo | 36.28 | **29.54** | **32.50** | 28.43 | 37.99 |
| Claude-3-Opus | 31.78 | 22.67 | 20.83 | 20.58 | 36.22 |
| Gemini-Pro | 21.13 | 15.50 | 15.35 | 9.57 | 13.33 |
| Baichuan-VL | 30.54 | 25.98 | 25.83 | 26.69 | 23.67 |
| Qwen-VL-Max | 28.71 | 21.86 | 28.33 | 20.86 | 19.09 |
| Qwen-VL-Plus | 16.70 | 17.07 | 18.67 | 16.67 | 11.46 |
| GLM4V | 8.94 | 12.57 | 5.13 | 7.32 | 10.55 |
| *Open-source Models* | | | | | |
| Yi-VL-34B | **16.78** | **15.84** | 7.02 | 9.79 | **11.44** |
| DeepSeek-V2 | 12.71 | 14.87 | 6.19 | **10.60** | 9.61 |
| LLaVA-1.6-7B | 9.76 | 8.58 | **15.79** | 3.57 | 10.77 |

Table 3: **Comparison of Models on different knowledge stages** (E: Easy, M: Medium, D: Difficult, Avg: Average Score)

| Model | Elementary | | | | Middle | | | | High | | | |
|-------|-----|-----|-----|-----|-----|-----|-----|-----|-----|-----|-----|-----|
| | avg | E | M | D | avg | E | M | D | avg | E | M | D |
| *Closed-source Models* | | | | | | | | | | | | |
| GPT-4V | 36.04 | 57.58 | 38.64 | 10.71 | **36.42** | 40.38 | **34.95** | 30.14 | **28.08** | 33.26 | 24.38 | **22.57** |
| GPT4-turbo | **37.71** | **72.73** | **38.79** | **18.33** | 35.12 | 37.22 | 34.51 | **30.44** | 26.06 | 28.65 | **25.19** | 18.83 |
| Claude-3-Opus | 28.30 | 33.33 | 31.10 | 10.04 | 31.04 | 31.29 | 33.97 | 12.22 | 19.17 | 24.07 | 16.41 | 15.15 |
| Gemini-Pro | 25.79 | 48.48 | 26.91 | 11.29 | 17.20 | 19.19 | 16.29 | 15.07 | 10.22 | 12.74 | 8.90 | 5.03 |
| Baichuan-VL | 29.85 | 35.00 | 31.45 | 18.33 | 29.96 | 28.94 | 32.57 | 21.38 | 22.33 | 27.59 | 17.42 | 16.01 |
| Qwen-VL-Max | 34.82 | 42.86 | 36.65 | 20.45 | 24.87 | 25.70 | 24.96 | 20.72 | 16.95 | 18.97 | 15.61 | 14.92 |
| Qwen-VL-Plus | 20.49 | 40.00 | 21.23 | 9.20 | 19.16 | 21.11 | 18.83 | 13.19 | 11.00 | 13.94 | 9.29 | 5.83 |
| GLM4V | 10.32 | 33.29 | 9.62 | 4.29 | 13.28 | 17.07 | 14.85 | 12.89 | 7.64 | 8.73 | 11.11 | 4.08 |
| *Open-source Models* | | | | | | | | | | | | |
| Yi-VL-34B | **14.99** | 40.00 | **16.13** | 3.32 | **16.38** | 16.31 | **17.10** | 11.67 | **12.14** | 11.65 | **12.96** | **10.58** |
| DeepSeek-V2 | 13.74 | **42.42** | 13.73 | 2.87 | 14.93 | 14.68 | 14.47 | **19.09** | 10.18 | 8.29 | 12.46 | 7.99 |
| LLaVA-1.6-7B | 9.77 | 35.21 | 10.82 | **7.12** | 10.37 | 9.79 | 10.90 | 9.07 | 7.57 | 8.41 | 6.53 | 4.54 |

educational stages. These models demonstrate superior accuracy, especially in structured question types like proof questions and in areas requiring advanced mathematical reasoning, such as algebra and probability. However, as the difficulty level increases, all models experience a decline in accuracy, with the most significant drops occurring between the middle and high school stages. GLM-4V performs particularly poorly at the high school level, highlighting the challenges that remain in achieving consistent performance on difficult math problems.

## 4.3 STABILITY RESULTS AND ANALYSIS

In this subsection, we perform a stability test for GPT4V, Claude-3-Opus, Baichuan-VL, and Qwen-VL-Max. We selected 300 problems and tested each model five times on each problem. The number of correct answers across these attempts was calculated to assess the stability of each model. As shown in Figure 6, none of the models demonstrate high stability—only about 25% of the problems were answered correctly in all five attempts. Therefore, it's imperative to focus on enhancing the stability and robustness of math MLLMs, as consistent performance across repeated trials is crucial for their practical application in real-world scenarios. This finding also suggests that future research should explore methods to reduce variability in model outputs, ensuring more reliable and trustworthy results.

## 4.4 ANSWER LENGTH AND ACCURACY

**Distribution of Answer Lengths**    From Figure 5, we observe distinct patterns in the distribution of answer lengths across different models. Notably, GPT-4V and Baichuan-VL tend to generate a larger proportion of shorter answers. As illustrated in Figure 7, it is evident that shorter but accurate

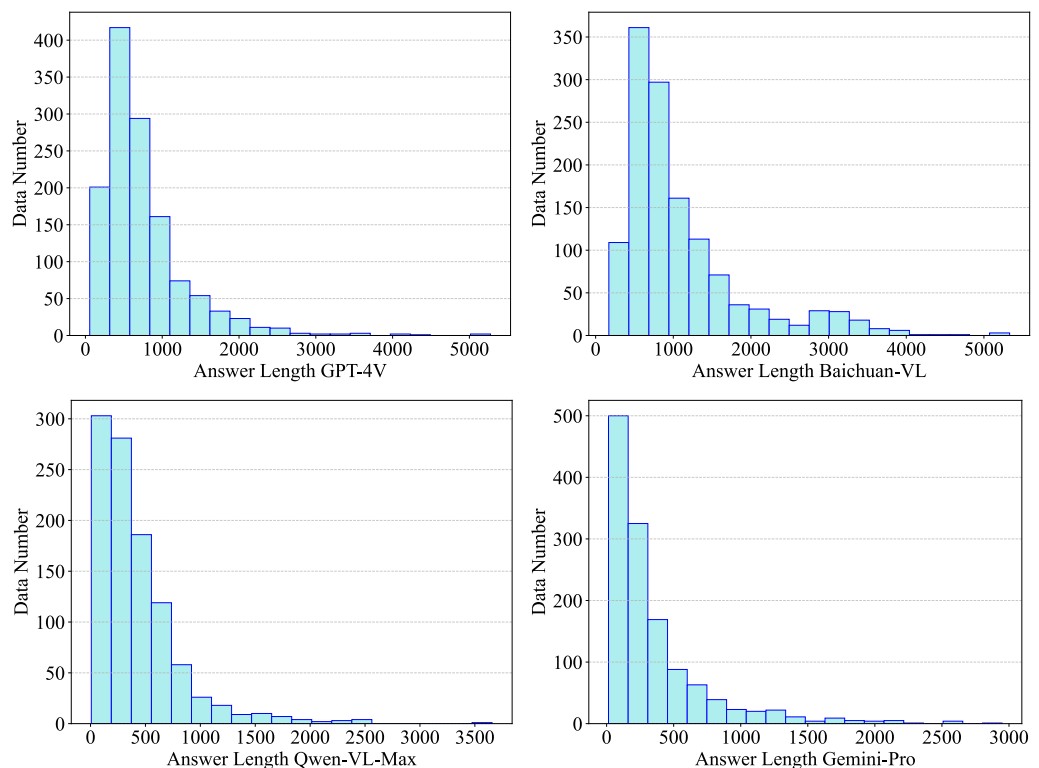

Figure 5: The variation of accuracy with answer length.

answers are more likely to achieve higher scores. This trend highlights the efficiency of models that can deliver concise and precise responses, particularly in scenarios where brevity is valued.

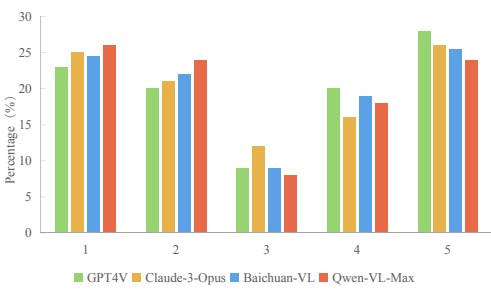

Figure 6: Stability Analysis: For each problem, the model is tested five times. The numbers 1 to 5 represent the proportion of correct responses.

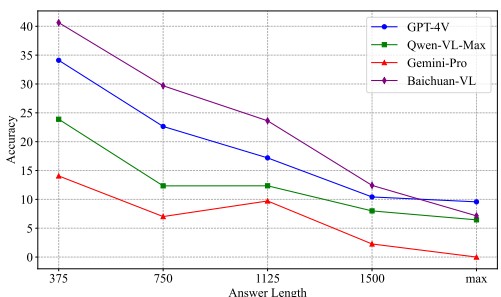

Figure 7: The variation of accuracy with answer length.

**Analysis of Answer's Length**  In our evaluation of the MathScape benchmark, we observed that there is no straightforward positive correlation between answer length and accuracy. In fact, as shown in Figure 7, when the length of the answer increases, the accuracy tends to decrease. This result demonstrates the robustness of the MathScape benchmark, ensuring that models cannot simply inflate their scores by producing longer answers. Such a design effectively prevents any biases in answering strategies, ensuring that the benchmark and evaluation method accurately reflects a model's true ability to understand and solve mathematical problems, rather than gaining an unfair advantage through verbose responses.

## 5 CHALLENGES AND FUTURE DIRECTIONS

As highlighted in Section 4, none of the models achieved strong performance on the MathScape benchmark. In this section, we present several case studies to illustrate the challenges faced by current MLLMs and propose potential future directions for enhancing their mathematical capabilities.

### 5.1 CHALLENGES

In this subsection, we explore the main reasons why models provide incorrect answers to image-text mathematical problems. These errors are mainly due to challenges in understanding and interpreting the information. We can break down these challenges into the following specific reasons:

**Unable to Retrieve Information from the Image:** This is one of the most common errors, where models may fail to extract all the relevant information from the image. For instance, when interpreting complex geometric patterns, it's easy to overlook certain data or conditions, leading to incorrect answers. As shown in Case Study 1 in Figure 8, the model provided an incorrect proof due to its incomplete understanding of the image.

**Misunderstanding of Graphic Positioning:** This issue involves the accurate understanding of the spatial layout of graphics. For instance, in geometry problems, errors can occur if the model fails to correctly recognize the lengths or angles of figures. Such mistakes often stem from a lack of deep understanding of graphic properties or insufficient ability to shift perspectives. In Figure 9, Case Study 2, the model incorrectly interprets the distance from point A to 0 as $\sqrt{2}$.

**Insufficient Reasoning Ability:** This issue arises from the limited logical reasoning capabilities of LLMs. Even when the image information is provided correctly, the LLM may still produce incorrect responses. As shown in Case Study 3 in Figure 10, the LLM fails to solve the complex problem correctly and makes errors in the process.

Overall, the challenges for multimodal large models primarily focus on the interpretation of visual information and the inherent reasoning abilities of the models.

### 5.2 FUTURE DIRECTIONS FOR MATH MLLMS

MathScape have introduced several challenges for MLLMs, as mentioned in section 5.1. In this section, we summarize future directions for MLLMs.

**Stronger LLMs** As outlined in Section 5.1, it is clear that LLMs exhibit limitations in mathematical reasoning. Moreover, all visual information must be processed by the LLM, further constraining its problem-solving capabilities. To enhance the mathematical reasoning proficiency of MLLMs, it is essential to develop more advanced LLMs with stronger mathematical reasoning capabilities.

**Better Pattern Recognition** Improving pattern recognition is essential for enhancing the performance of MLLMs, particularly in tasks involving complex visual information. Current models often struggle with identifying and interpreting intricate patterns in images, such as geometric configurations, charts, and fine-grained visual details. Future research should focus on developing models that can more accurately recognize and differentiate patterns, especially when they are complex.

## 6 CONCLUSION

Recently, MLLMs have emerged as powerful models for answering questions across multiple domains. However, comprehensive benchmarks that reflect real-world scenarios are needed to evaluate their mathematical performance. In this paper, we introduce MathScape, a new benchmark designed to assess the math capabilities of MLLMs using entirely original, leak-free images. Additionally, we propose a novel two-step evaluation method specifically for assessing long answers. MathScape not only challenges existing MLLMs but also aims to inspire the development of more advanced math-focused MLLMs.

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

OUTLINE

The appendix is organized as follows:

**A. Additional Case Studies** In our case study analysis, we presented three examples that highlight the challenges VLLMs face in solving MathScape problems. These findings underscore the need for more advanced VLLMs capable of tackling mathematical tasks effectively.

**B. Visualization of MathScape** In this section, we provide visualizations of the MathScape dataset to offer a clearer and more direct understanding of the benchmark's structure and content.

**C. Prompts for Inference, Answer Extraction, and Scoring** We have summarized the prompts to more clearly demonstrate the process of dataset inference and evaluation, ensuring better reproducibility of the results.

# A  CASE STUDY

We present case studies of model responses, highlighting challenges and future directions. As illustrated in Figures 8, 9, and 10, the model faces difficulties with complex reasoning tasks, underscoring the need for more robust reasoning models.

Given the figure, it is known that AM∥BN, ∠A=60∘, point P is a moving point on the ray AM (not coincident with A), BC and BD bisect ∠ABP and ∠PBN respectively. The ray AM intersects C and D. (There needs to be a reasoning process, but it is not necessary to write out every step.)

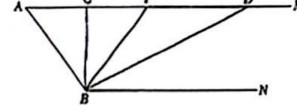

1. Find the measure of ∠CBD
2. Prove that ∠APB=2∠ADB
3. When point P moves such that ∠ACB=∠ABD, find the measure of ∠ABC.

**(a) Math Problem**

First, since BD is the bisector of ∠PBN, similarly, we can conclude that ∠PBD=∠ABD=60°. Then, because ∠APB=∠ABP+∠ABD, we know that ∠ABP=∠ABC (since BC bisects ∠ABP), which equals 60°. Therefore, ∠APB=60°+60°=120°. Now, we can calculate ∠ADB. Since ∠ABD=60°, and we also know that ∠BAD=60°, in the isosceles triangle ∆ABD, ∠ADB=180°−∠BAD−∠ABD=180°−60°−60°=60∘ Therefore, we can say ∠APB=120°=2×60°=2∠ADB

**(b) GPT4 Response**

**Judge:** The proof is complete incorrect.
**Difficulty:** Recognition of spatial angle relationships, spatial deduction and proof.

**(c) Challenges**

Figure 8: Illustration of Case Study 1. The red sign marks the incorrect part of the answer.

As shown in the figure, using a unit length segment on the number line as the side of a square, with the point representing the number 1 as the center of a circle, a semicircle is drawn with a radius equal to the side length of the square. The semicircle intersects the number line at point A. The number represented by point A is _____.

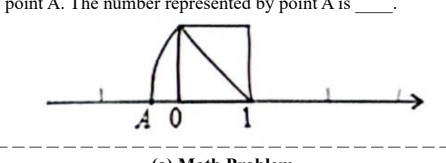

**(a) Math Problem**

**Judge:** The answer is incorrect.
**Difficulty:** Weak spatial ability and spatial position reasoning skills.

**(c) Challenges**

First, the problem describes a square with a side length equal to the unit length representing the number 1 on the number line. Let's denote the side length of this square as 1 unit. Next, the problem states that the point representing the number 1 is the center of a circle, and the radius of the arc is equal to the length of the diagonal of the square. We know that the length of the diagonal of a square is$\sqrt{2}$ times the side length, so in this case, the diagonal length is $\sqrt{2}$. Therefore, with 1 as the center and $\sqrt{2}$ as the radius, the length of this arc represents the distance on the number line. Simply put, the arc intersects the number line at point A, and the distance from point A to the origin (0) on the number line is $\sqrt{2}$.
So, the number represented by point A is 1 plus $\sqrt{2}$, which means point A represents the number 1+ $\sqrt{2}$.
**Final Answer:** 1+ $\sqrt{2}$

**(b) GPT4 Response**

Figure 9: Illustration of Case Study 2. The red sign marks the incorrect part of the answer.

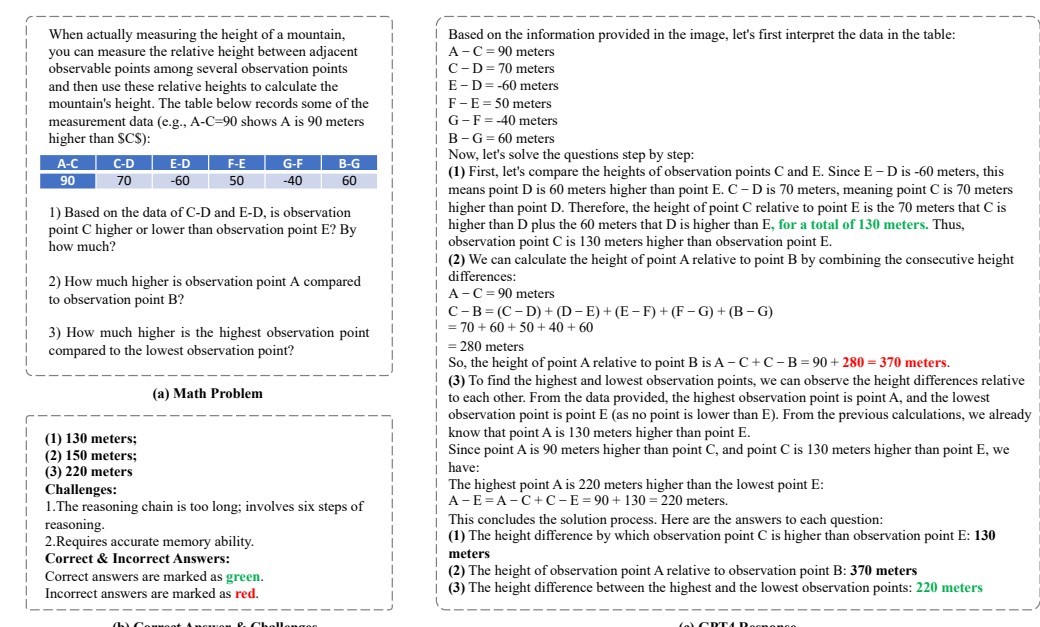

**(a) Math Problem**

When actually measuring the height of a mountain, you can measure the relative height between adjacent observable points among several observation points and then use these relative heights to calculate the mountain's height. The table below records some of the measurement data (e.g., A-C=90 shows A is 90 meters higher than $C$)

| A-C | C-D | E-D | F-E | G-F | B-G |
|---|---|---|---|---|---|
| 90 | 70 | -60 | 50 | -40 | 60 |

1) Based on the data of C-D and E-D, is observation point C higher or lower than observation point E? By how much?

2) How much higher is observation point A compared to observation point B?

3) How much higher is the highest observation point compared to the lowest observation point?

**(b) Correct Answer & Challenges**

(1) 130 meters;
(2) 150 meters;
(3) 220 meters
**Challenges:**
1.The reasoning chain is too long; involves six steps of reasoning.
2.Requires accurate memory ability.
**Correct & Incorrect Answers:**
Correct answers are marked as **green**.
Incorrect answers are marked as **red**.

**(c) GPT4 Response**

Based on the information provided in the image, let's first interpret the data in the table:
A − C = 90 meters
C − D = 70 meters
E − D = -60 meters
F − E = 50 meters
G − F = -40 meters
B − G = 60 meters
Now, let's solve the questions step by step:
**(1)** First, let's compare the heights of observation points C and E. Since E − D is -60 meters, this means point D is 60 meters higher than point E. C − D is 70 meters, meaning point C is 70 meters higher than point D. Therefore, the height of point C relative to point E is the 70 meters that C is higher than D plus the 60 meters that D is higher than E, **for a total of 130 meters.** Thus, observation point C is 130 meters higher than observation point E.
**(2)** We can calculate the height of point A relative to point B by combining the consecutive height differences:
A − C = 90 meters
C − B = (C − D) + (D − E) + (E − F) + (F − G) + (B − G)
= 70 + 60 + 50 + 40 + 60
= 280 meters
So, the height of point A relative to point B is A − C + C − B = 90 + **280 = 370 meters**.
**(3)** To find the highest and lowest observation points, we can observe the height differences relative to each other. From the data provided, the highest observation point is point A, and the lowest observation point is point E (as no point is lower than E). From the previous calculations, we already know that point A is 130 meters higher than point E.
Since point A is 90 meters higher than point C, and point C is 130 meters higher than point E, we have:
The highest point A is 220 meters higher than the lowest point E:
A − E = A − C + C − E = 90 + 130 = 220 meters.
This concludes the solution process. Here are the answers to each question:
**(1)** The height difference by which observation point C is higher than observation point E: **130 meters**
**(2)** The height of observation point A relative to observation point B: **370 meters**
**(3)** The height difference between the highest and the lowest observation points: **220 meters**

Figure 10: Illustration of Case Study 3. The red sign marks the incorrect part of the answer.

# B   VISUALIZATION OF MATHSCAPE

We include additional math samples in MathScape, translated into English, as shown in Figure 11. Furthermore, we provide examples of human-captured photos within the MathScape dataset.

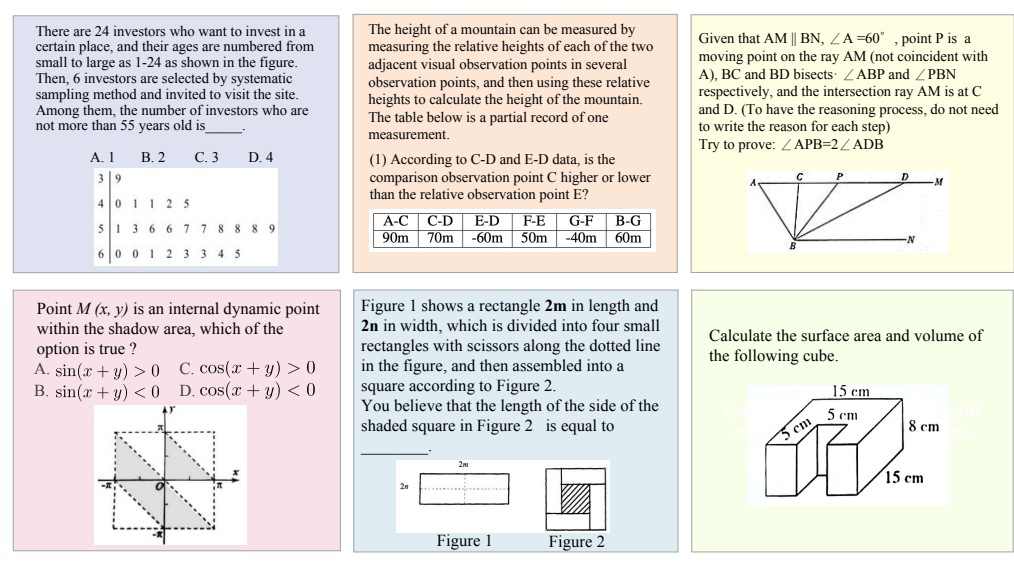

Figure 11: Math problem samples in MathScape.

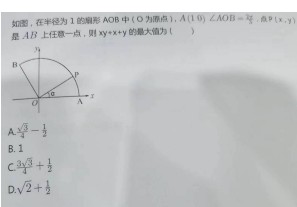 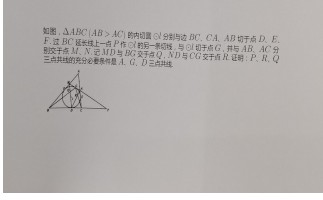 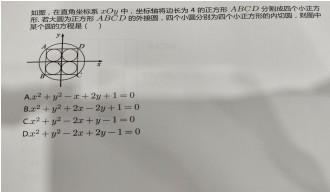

Math Images Type 1: Print & Take Photo

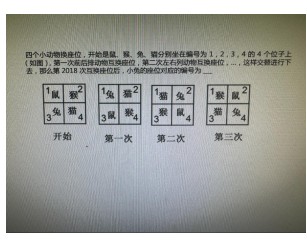 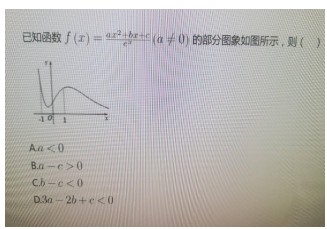 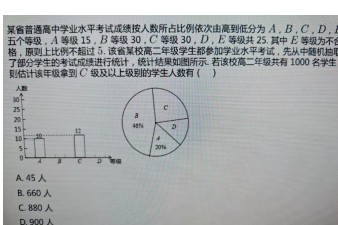

Math Images Type 2: Take Photo of the Screen

Figure 12: Math problem real photo samples in MathScape.

## C    PROMPT FOR INFERENCE, EXTRACTING AND SCORING ANSWERS

We summarize the prompt for scoring answers in Figure 14.

**System:** "You will play the role of a problem-solving assistant skilled in solving math problems. Your task is to analyze and solve math problems based on both textual and visual information. You need to understand the meaning of the problem presented in the image and combine the text recognized from the image to solve the problem step by step."

**Demand:** "You need to have a comprehensive understanding of both the text and the image, and then answer the question in the text.

**Note:** The final output should be in JSON format, with the following structure: { "solution": "Explanation of the problem-solving process..." , "answer": "Final answer" }."

*Prompt-Inference*

You need to extract the expressions of the student's answers for each sub-question.
Student's response: {response}
You need to output the following:
Student's answers: {{Extracted student's answers result:
(1){{Student's answer}}
(2){{Student's answer}}
(3){{Student's answer}}
(4)....}}                                    *Prompt-Extract*

Figure 13: Prompts for inference and extracting answers.

Task Description: Evaluate whether the student's answer to the given math problem is correct.

Input:
1. Problem Description: [Detailed description of the problem, including necessary mathematical formulas and conditions.]{question},
2. Reference Answer: [Detailed explanation of the correct answer, including the calculation process and result.]{answer},
3. Student's Answer: [The student's provided answer, including the calculation process and result.]{response},

Requirements:
- Carefully compare the student's answer with the reference answer.
- Analyze the correctness of the student's answer, including the calculation process and the final result.
- If the student's answer is incorrect, identify the error and briefly explain the reason for the mistake.
- Provide a concise evaluation conclusion, clearly stating whether the student's answer is correct.

Example:
Problem Description: Calculate the area of a triangle with a base of 6 cm and a height of 3 cm.
Reference Answer: (1) Area = 0.5 * base * height = 0.5 * 6 cm * 3 cm = 9 cm².
Student's Answer: (1) Area = 6 cm * 3 cm = 18 cm².

Evaluation:
(1) False, explanation as follows:
- The student's calculation process ignored the 1/2 coefficient in the area formula.
- The result is incorrect; the correct calculation should yield 9 cm², not 18 cm².
- Conclusion: The student's answer is incorrect.

Based on the above task description and requirements, compare the reference answer and the student's answer in order. Carefully consider whether they are consistent.
2. If the student's answer is correct, output True; otherwise, output False and provide an evaluation conclusion.

You need to output:
Only the True or False for each question, example: Judgement result: (1) True, (2) False, (3) True
Explanation as follows: (1)... (2)... (3)...

Figure 14: Prompt used for scoring answers.

