# OpenReview forum: "MathScape: Evaluating MLLMs in Multi-modal Math Scenarios through a Hierarchical Benchmark"
_ICLR.cc/2025/Conference — ICLR 2025 Conference Withdrawn Submission_

### Official Review · Reviewer_aDbf · 2024-10-25

**Soundness:** 2
**Presentation:** 2
**Contribution:** 2
**Rating:** 3
**Confidence:** 4

**Summary:**

The paper introduces a new benchmark termed MathScape for assessing the capabilities of Multimodal Large Language Models (MLLMS) in solving mathematical problems that involve both visual and textual information. MathScape addresses the gap in existing benchmarks by offering a more realistic testing environment with image-based math problems. The benchmark is designed to evaluate the theoretical understanding and application ability of MLLMS through a categorical hierarchical approach. Finally, the paper reports on a multi-dimensional evaluation of 11 advanced MLLMS, revealing the challenges posed by the benchmark and identifying current limitations of these models.

**Strengths:**

1.Originality: The paper presents MathScape, an innovative benchmark that combines real-world math problems captured in images with their correct answers, closely mirroring real-world scenarios and providing a more comprehensive assessment of MLLMS.

2.Quality: The benchmark covers a wide range of difficulty levels, question types, and knowledge areas, which is commendable.

3.Clarity: The paper is structured with clear explanations of the benchmark construction process, evaluation approach, and results.

**Weaknesses:**

1. I think authors should be aware that except for those previous works you mentioned, there are many other mathematical reasoning benchmarks this year [1,2,3,4,5], especially with a similar focus on multimodal reasoning. Hence, two of your contributions (New Persepective and New Benchmark) may lack novelty. Besides, New Method (i.e., how you construct and evaluate) is a fair but not strong contribution to the MLLM community.

2. The paper indicates that the dataset primarily consists of Chinese problems. I think this will narrow the contribution as well. Besides, educational levels (i.e., primary/middle/high school) are highly different between China and Western countries. So it is better if you can address this limitation, such as including a comparison of educational standards or proposing how the benchmark could be adapted for different educational systems.

3. The analysis is not sufficient for benchmark work. For example, we need to know the proportion of diverse reasons why the best model provides incorrect answers (e.g., failure to retrieve the visual information; misunderstanding of positioning; etc.) in both the whole dataset and each dimension. Furthermore, more bad cases are needed.

4.The evaluation focuses on a set of state-of-the-art models, but it might be beneficial to include GPT-4o, which has been proven for its effectiveness for complex reasoning. Besides, math-specific MLLMs should be included as well, since you also mentioned them in your related works.

References:

[1] We-Math: Does Your Large Multimodal Model Achieve Human-like Mathematical Reasoning?

[2] IsoBench: Benchmarking Multimodal Foundation Models on Isomorphic Representations

[3] CMM-Math: A Chinese Multimodal Math Dataset To Evaluate and Enhance the Mathematics Reasoning of Large Multimodal Models

[4] CMMaTH: A Chinese Multi-modal Math Skill Evaluation Benchmark for Foundation Models

[5] ErrorRadar: Benchmarking Complex Mathematical Reasoning of Multimodal Large Language Models Via Error Detection

**Questions:**

1. Based on Weakness 1, please elaborate on the most significant contribution of this benchmark, compared to existing multimodal math reasoning benchmarks. You can ignore the parallel research, but I think the related work is not comprehensive yet.

2. I think some of current MLLMs may suffer from different lingual contexts. Therefore, is it possible to expand your work to English problems, or explore the performance difference between Chinese and English.

3. The evaluation part should include GPT-4o if possible. Besides, it should dive deeper into analysis of bad case category proportions and more bad case analysis.

4. I wonder if geometric problems are the hardest type, as it also needs a more complex visual perception of specific components such as angles and lines.

5. The performance tables need to include parameter size for each open-source models. Also, a scaling analysis is needed if possible.

---

### Official Review · Reviewer_ajb2 · 2024-10-31

**Soundness:** 3
**Presentation:** 3
**Contribution:** 2
**Rating:** 5
**Confidence:** 4

**Summary:**

This paper proposes MathScape, a new benchmark for multimodal math problems to evaluate the related capabilities of MLLMs. The collected datasets contain images with both math figures and questions. The author also uses a two-step evaluation method to first extract answer and then judge the correctness using LLMs. The author evaluate different MLLMs on this new benchmark with detailed analysis.

**Strengths:**

1. The data collection process is delicate and clearly stated with clear figures.

2. The classification process of math problems are well defined and reasonable.

3. The author provide detailed analysis of accuracy and answer length. This provides some insights to future math MLLMs.

**Weaknesses:**

1.  The contribution of this paper is overclaimed. To the best knowledge, MathVerse contains six versions of a problem and the 'vision-only' one also contains both math figures and question in the image, similar to the contribution of this paper.

2. The two-step evaluation cannot be viewed as an important contribution, since MathVista also uses an LLM (ChatGPT) to extract answers from the free-form response of models as the first evaluation stage.

3. The evaluation of some math-domain MLLMs is missing on MathScape, for example: G-LLaVA and Math-LLaVA.

4. Human performance is needed on MathSacpe for better reference.

**Questions:**

More visualization results of the evaluation process can also help to understand the proposed evaluation strategy.

---

### Official Review · Reviewer_4Gu1 · 2024-11-03

**Soundness:** 3
**Presentation:** 3
**Contribution:** 2
**Rating:** 6
**Confidence:** 4

**Summary:**

The paper introduces MathScape, a new benchmark that evaluates the mathematical capabilities of Multimodal Large Language Models using photo-based math problems. Unlike previous benchmarks, MathScape integrates problem statements and visual elements within a single image using a print-and-photo or display-and-photo approach. The authors collected 1,325 images of school-level mathematical problems in multiple choice, free-form, and proof formats (38%, 56%, and 5% respectively). They evaluated 11 closed and open-weight Large Language Models and provided a case study. The results demonstrate that MathScape is challenging even for state-of-the-art models, particularly in the stage of extracting problem statement from image input.

**Strengths:**

* Benchmark Size and Coverage: The dataset covers a wide range of topics and difficulty levels; 1.2k samples allow for a statistically significant assessment of MLLMs in each subject (except for equations).
* Data Quality Control: Post-photo quality control and classification is great addition, allowing reviewers to filter unreadable inputs.
* Evaluation Approach: The two-step evaluation method with sub-task scoring might reduce judgment errors and allows for more fine-grained analysis of the evaluation results.

**Weaknesses:**

* Insufficient Dataset Details: More comprehensive information about the dataset’s creation, sources, human annotators education level and potential biases would strengthen the paper.
* Limited Language Scope: The focus on Chinese problems limits the applicability of the benchmark to other languages and educational contexts. (Please clearly state the language scope in the abstract and/or in the introduction).
* Evaluation Method Reliance on LLMs: Using LLMs for scoring may introduce biases, as these models may share similar limitations with the models being evaluated. The judgment error is not addressed in the paper's results or case study.
* Lack of Comparative Analysis: Given that all of the problems are available in textual format, the paper will benefit from including correlation analysis between original problems and photo-converted problems solve rate.

**Questions:**

* Evaluation Method Validation: How does the proposed two-step evaluation method compare with traditional evaluation methods in terms of reliability and validity?
* Token Limit Impact: How does the 2048-tokens generation limit affect the results, especially for verbose models? What percentage of responses are truncated by this limit?
* JSON format output: The constraining model to output JSON format is known to decrease the quality of the generated content (e.g. https://arxiv.org/abs/2408.02442v1). Why the authors choose to stick to this method? What is the impact of such format constrains in current settings?

**Details Of Ethics Concerns:**

* Problem Sources and Copyright: The problems are stated to be collected from school exams and homework, which raises questions about the original sources and copyright status of these data samples.
* Fair Compensation: The dataset collection process involved human reviewers for quality control, but it is unclear whether these reviewers received fair compensation for their work.

---

### Official Review · Reviewer_pYVk · 2024-11-04

**Soundness:** 2
**Presentation:** 1
**Contribution:** 2
**Rating:** 3
**Confidence:** 4

**Summary:**

This paper proposed a new multimodal mathematical evaluation benchmark called MathScape, which consists of 1325 problems. MathScape combines both figures and mathematical text descriptions into images, which presents a challenge to multimodal large language models. This paper also introduced a two-stage evaluation method to evaluate long responses to math questions. They tested several MLLMs in different data-splitting methods to show results from different perspectives.

**Strengths:**

1. MathScape contains 1325 high-quality human-collected real multimodal mathematical problems.
2. Authors conduct an analysis of the relationship between answer length and performance, which is interesting.

**Weaknesses:**

1. In the first challenge, the author said no existing benchmarks have both the mathematical description and figures being captured together in a single image. However, in MathVerse, one category of questions does provide descriptions and figures together. For the second challenge, the author claims existing method cannot assess long-form responses. But MathVerse proposes a method to assess the correctness of each step of a chain-of-thought response. Authors should conduct a more comprehensive literature review of the multimodal mathematical evaluation domain.
2. This paper is not well organized and written. This means it is not easy to read and understand. For example, section 3.1 is oversimplified. The authors did not mention where they collected the mathematics question and what is the original format of the question documents. Besides, it’s not clear what kinds of annotations are done. What is “knowledge-based classification”?
3. The proposed two-step evaluation method heavily relies on LLM’s ability to decompose and judge the answer. This may cause some errors in the progress. Did the authors examine how accurate LLMs are on each of the evaluation tasks?
4. For the evaluation part:
    1. The model “GLM4V” is without citation, and it is an open-sourced model from my knowledge. (https://huggingface.co/THUDM/glm-4v-9b). Besides, the open-source models in Line 278 are not cited properly. These kinds of format errors cause the paper to be hard to read.
    2. Some reference performance is not provided: e.g., frequent choice, random choice, and human performance.
    3. DeepSeekV2 is not in the evaluation setup models, did you mean DeepSeek-VL?
    4. The performance on proof questions is higher than on choice and solution questions. This is uncommon and the reason given by the authors is not convincing. They said, “The structured format and clear information in proof questions make them easier”. However, when testing models on different kinds of questions. The format of questions is supposed to be similar unless the question format (structure or non-structure) is the primary research topic.
    5. The authors provide limited insights of the performance on MathScape. Results such as “the closed-source models are more accurate than open-source ones” reveal little information.
5. MathScape claims that it is the first to combine both figures and mathematical text descriptions in a single image. What unique challenge does this format of data bring to models? Did the authors dive deep into analyzing the different challenges present by MathScape and other multimodal mathematical benchmarks?

**Questions:**

Please see above.

---

### Note · Authors · 2024-11-21

I have read and agree with the venue's withdrawal policy on behalf of myself and my co-authors.